# Remediation of Acid Mine Drainage (AMD) Using Steel Slag: Mechanism of the Alkalinity Decayed Process

**DOI:** 10.3390/ijerph20042805

**Published:** 2023-02-04

**Authors:** Lei Yang, Yuegang Tang, Duanning Cao, Mingyuan Yang

**Affiliations:** 1School of Geosciences and Surveying Engineering, China University of Mining and Technology, Beijing 100083, China; 2Guizhou Zhonggui Environmental Technology Co., Ltd., Guiyang 550008, China; 3School of Chemistry and Chemical Engineering, Chongqing University of Technology, Chongqing 400054, China

**Keywords:** acid mine drainage, steel slag, acid neutralization capacity, alkaline

## Abstract

Steel slag has been proven to be an effective environment remediation media for acid neutralization, and a potential aid to mitigate acid mine drainage (AMD). Yet its acid neutralization capacity (ANC) is frequently inhibited by precipitate after a period of time, while the characteristics of the precipitate formation process are unclear yet. In this study, ANC for basic oxygen steel slag was conducted by neutralization experiments with dilute sulfuric acid (0.1 M) and real AMD. Some partially neutralized steel slag samples were determined by X-ray diffraction (XRD), scanning electron microscopy combined with an energy dispersive spectrometer (SEM-EDS), and N_2_ adsorption tests to investigate the potential formation process of the precipitate. The results indicated that Ca-bearing constitutes leaching and sulfate formation were two main reactions throughout the neutralization process. A prominent transition turning point from leaching to precipitate was at about 40% of the neutralization process. Tricalcium silicate (Ca_3_SiO_5_) played a dominant role in the alkalinity-releasing stage among Ca-bearing components, while the new-formed well crystalline CaSO_4_ changed the microstructure of steel slag and further hindered alkaline components releasing. For steel slag of 200 mesh size, the ANC value for the steel slag sample was 8.23 mmol H^+^/g when dilute sulfate acid was used. Neutralization experiments conducted by real AMD confirmed that the steel slag ANC was also influenced by the high contaminants, such as Fe^2+^, due to the hydroxides precipitate reactions except for sulfate formation reactions.

## 1. Introduction

Acid mine drainage was formed from the sulfide minerals in the earth during mining activities [1]. Once the sulfide contained in minerals was exposed to water and oxygen, most sulfide minerals were liable to oxidize to produce acid, metal ions, and sulfate, which would enter groundwater and surface water [2]. For typical AMD from closed coal mines, its pH generally ranged from 2 to 5, with a high ferrous concentration of 10–1000 mg/L and other trace metals, including Pb^2+^, Mn^2+^, Zn^2+,^ and so on. On the other hand, to reduce the overcapacity in the coal industry and protect the mining environment, many small-scale mines were forced to stop production recently, especially in China. Although the relevant coal mines were closed, the AMD formed underground, would effluent persistently, with low pH and high concentration of trace toxic elements, which need remediation [3,4,5,6,7].

As the largest crude steel production country, China’s steel slag utilization ratio was only about 30%, which is much lower than that of the developed countries. The present utilization ratios in various applications for China and developed countries were listed in Appendix A [8]. It can be seen that road project construction, cement production and other construction materials were the primary utilization applications currently [2,9]. Although many efforts had been made to increase the steel slag utilization ratio as construction materials, the performance of used construction materials was frequently substandard [10], because free CaO (f-CaO), free MgO (f-MgO) and other easily hydrated components could deteriorate the cement stability associated with the volume stability [5,11]. When steel slag was used in unbound and bitumen-bound layers, its acceptable contents of f-CaO and f-MgO should be lower than 7% and 4% [5]. Therefore, it is necessary to promote the steel slag utilization ratios alternatively, such as soil remediation [12], water treatment [13,14,15,16,17], AMD neutralization [9,10] and other promising pollution control applications.

Previous studies have confirmed that steel slags were more excellent in alkaline production than traditional alkaline materials, such as limestone [1,18,19], lime, soda alkaline, and dolomite [20,21]. Therefore, when the steel slags were employed in AMD treatment, acidity in wastewater was neutralized by alkaline components in steel slags, such as f-CaO, f-MgO, Ca_2_SiO_4_ (C_2_S), Ca_3_SiO_5_ (C_3_S), CaFe_2_O_4_ (C_2_F) [22]. It was found that the alkalinity of limestone leach bed was about 5 mg/L as CaCO_3,_ with a maximums value of 80 mg/L as CaCO_3_, while the average alkalinity concentration of steel slag leaching beds was above 100 mg/L as CaCO_3_ [22,23]. Another superiority of steel slag was its persistence in alkalinity releasing time of high concentrations because steel slag does not absorb CO_2_ from the air and convert back to relatively insoluble calcite [22,23,24]. It is an extremely important property for AMD remediation since the high levels of alkalinity could last for many years. However, using steel slag to treat AMD was not a widespread application in practice. As reported, it was currently used in the area of West Virginia, U.S., and South Africa in the form of steel slag leach bed (SSLB) [23,25]. To the authors’ knowledge, remediation of AMD by steel slag has not been attempted in China.

Although the steel slag does not react with CO_2_ in air, its alkalinity releasing behaviors was influenced by many factors during the treatment process. Performance of operated steel slag leach bed (SSLB) showed that the ANC of steel slag were influenced by water quality [23], slag alkalinity loading [26], slag particle size distribution [27], etc. Compared the SSLB built for one year and historical bed operated for many years, it was found that the precipitates in the effluent piping of the SSLB and internal of slags reduced the ANC of the slag resulting in a lower effluent pH than the designed value [14,23].

To ensure acid mine effluent meet the designed value or emission standard, it is of significance to guarantee steel slags in SSLB have sufficient alkalinity to neutralize acidity in AMD, however, it is extremely difficult to monitor the alkalinity attenuation process in real AMD remediation process. In this study, acid neutralization capacity (ANC) was employed as a measurable indicator to express the ability to neutralize the acidity in AMD for steel slags. It is known that ANC gradually decayed with the treatment time extended, however, it was influenced by many factors, such as steel slag composition, alkaline component leach-ability, various chemical reactions, interfaced changes caused by the reaction, and so on. Therefore, the steel slag ANC variation is a complicated process, and it is important to find the potential ANC change rule and critical factors in the neutralization process. The main objective of the manuscript was the study the decay mechanism of the acid neutralization capacity for the steel slag during the acid mine drainage remediation process aiming to elucidate the precipitate formation process and to provide an operational guide for AMD remediation using steel slag.

## 2. Experiments

### 2.1. Materials

Basic oxygen furnace steel slag from Shougang Group Shuicheng Steel Co., Ltd. (Liupanshui, Guizhou Province, China) was used in the study. Before use, raw steel slag was dried overnight at 105 °C, ball milled, and screened through sieves of 20 mesh, 50 mesh, 100 mesh, and 200 mesh.

Both dilute sulfuric acid and real AMD samples were used to test the steel slag ANC, the concentration of the dilute sulfuric acid was 0.1 M. Real AMD was collected from a closed coal mine in Guizhou Province, China.

### 2.2. Experimental Procedure

#### 2.2.1. Alkalinity Production Determination

The term alkalinity production refers to the alkaline minerals in leaching solution released from steel slags, which was determined according to the Chinese national standard method (SL83-1994). As the standard method described, 1.0 g steel slag sample of 200 mesh size was mixed with 100 mL deionized water under a rotating speed of 100 r/min for preset time. Filtrate was separated using a 0.45 µm filter (Merck, Lowe, NJ, USA), then its pH was determined while the alkalinity was calculated and expressed as calcium carbonate (CaCO_3_).

#### 2.2.2. ANC Determination

ANC determination process was conducted by typical titration method, in which AMD samples from automatic titrator were used to simulate inflow into slag leaching beds, while the flask containing slags was used as a neutralization reactor with its pH monitored, which was shown in Figure 1. Before titration procedure, 1.0 g and 100 mL DI water was mixed and stirred at 500 rpm for 4 h. During titration process, solution pH was determined before and after each titration procedure. It was found that when simulated AMD or real AMD added into solution, it was depleted immediately by the alkaline components in solution, causing solution pH had a re-rise value. With the added AMD increasing, the re-rise value became smaller since the remained alkaline components in slag was depleted gradually. Therefore, the neutralization endpoint was regulated at the time when the re-raised pH value was less than 0.05 after a titration procedure, i.e., the alkaline components in the steel slag were completely neutralized.

#### 2.2.3. Leaching Ratio

Leach ability of slag components was measured by leaching ratios, which were calculated according to following formula.
(1)$$r=\frac{c_{1}\cdot V}{c_{2}\cdot m}$$
where, r is leaching ratio for specific components; $c_{1}$ is concentration in the leach liquor; V is volume of filtrate; $c_{2}$ is mass percentage in slag samples; m is mass of the slag samples.

#### 2.2.4. Characterization Techniques

The chemical compositions in raw steel slag samples were quantitatively analyzed by X-ray fluorescence (XRF-1800). f-CaO in steel slag was determined according to acid titration method. Phase compositions were performed by powder X-ray diffraction (XRD) using a Smartlab X-ray diffractometer with the operating parameters of Cu Kα radiation (λ = 1.5418 Å) and 40 kV/200 mA power generator. An angular range of 10−90° 2θ was measured with a step size of 0.02° and a 1 s counting time per step. Identifications of all crystalline phases were undertaken with JADE6 software and the PDF-2 2004 database. Morphology of steel slag samples was observed using scanning electron microscopy coupled with energy dispersive spectrometer (SEM-EDS) to analyze the grain structure and mineralogy. All steel slag samples were sputtered coated with an approximately 5 nm thick layer of carbon before observation. Specific surface areas, average pore size, and pore volume of the steel slag samples were conducted by the Brunauer–Emmett–Teller method through N_2_ adsorption at 77 K, and pore distribution was calculated by the Bearrett–Joyner–Halenda (BJH) method. Composition Ca, Mg, Si, Al, Mn, Fe, and Cr in aqueous solutions were determined by inductively coupled plasma optical emission spectrometer (ICP-OES, Perkin Elmer Avio 500).

## 3. Results and Discussion

### 3.1. Characterization and Alkalinity Production

Except for the main chemical components in steel slags, shown in Table 1, there were also some trace ingredients for different steel slags, such as ZnO, SrO, MoO_3_, CuO, they made a negligible contribution to ANC because of their trace amount. Therefore, only constituents containing Ca, Fe, Si, Mg, Al, Mn, and Cr were considered in the experiment.

Steel slag is a heterogeneous material containing many crystalline phases. Diffraction lines in the XRD result revealed that major phases in the present sample contained dicalcium silicate (C_2_S, 2CaO·SiO_2_), tricalcium silicate (C_3_S, 3CaO·SiO_2_), calcium ferrite (C_2_F, 2CaO·Fe_2_O_3_), free CaO (f-CaO), and RO phase (a solid solution of CaO-FeO-MnO-MgO) as shown in Figure 2a, of which the content of free CaO in raw steel slag was 2.3%. The SEM results indicated that the microstructure and morphology of the raw steel slag samples had a porous surface, as shown in Figure 2b. Additionally, the alkalinity production test results in Figure 2c suggested that the alkalinity of the raw sample was in a range of 100–110 mg/L as CaCO_3_, and its corresponding pH was 10.5, which was consistent with previous results [23], however, the f-CaO content was not as high as the samples reported. It is known that the calcium-containing components in basic oxygen furnace slags mainly originated from the added flux materials during the steel-making process, such as lime and limestone [10]. The unreacted limestone left in steel slag existed in f-CaO form, while the reacted limestone was transferred to C_2_S, C_3_S, CF, and other minerals by molten reactions [25]. In other words, the f-CaO component in steel slag did not play a dominant role in the alkaline production process, but its porous microstructure provided a big surface area and inner channels which had a positive influence on ANC property [2].

### 3.2. Acid Neutralization Capacity Tests

Experimental results of the acid neutralization capacity test by simulated AMD (0.1 M H_2_SO_4_) are presented in Figure 3a. It can be seen that with the simulated AMD amount increasing, the pH of the solution decreased directly from an initial 10.5 to 2.15 at the endpoint gradually. It is known that the alkaline ingredients in steel slag could leach into the solution, raising the solution pH to the alkaline range. For the 1 g slag sample, the final pH was 10.5. While with the added dilution sulfuric acid amount increasing, the produced OH^−^ was neutralized, which make the pH increase slowly although the leaching reaction was going on. The total consumed volume of the simulated AMD was 115 mL for 1 g steel slag samples with an average particle size of 200 mesh. In addition, it was about 40 mL when the pH reached the neutral value of 7. The calculated ANC value was 8.23 mmol H^+^/g for the steel slag sample, which agreed well with the reported ANC results of the electric arc furnace steel slag. The electric arc furnace steel slag was neutralized by a similar method using HNO_3_ (1 M) as simulated AMD, when pH decreased to 7 [28], the calculated ANC of EAF slag was about 5.5 mmol H^+^/g slag and 7.5 mmol H^+^/g slag for short and long experiments, respectively.

Another significant factor on steel slag ANC property was particles size distribution, because smaller particles had larger surface area and are easier to release the alkaline components [27]. ANC results for slag sample of 20, 50, and 100 mesh size indicated that slag of bigger size had smaller ANC values, as shown in Figure 3b. For example, it was 8.53 mL simulated AMD for sample of 20 mesh size when reaching the endpoint. It was much less than the sample of 200 mesh size. It was reported that particles less than 3 mm were the most effective in AMD treatment because of rapid dissolution due to their high surface area [15]. However, fine particle slags in SSLB were more likely to precipitate on the bottom of beds, and to increase running resistance [27,29], especially for passive remediation. Furthermore, there is a severe problem when fine slag particles were used because the smaller slag particles are liable to release the heavy metals, especially EAF steel slag [27]. Therefore, it is of great significance to optimize the slag size distribution according to the ANC demand and the geographical environment of remediation site, so as to balance the advantages and disadvantages of the fine particle size.

### 3.3. Slag Characteristics Variation during Alkalinity Consumption Process

To investigate the steel slag changes during the titration process, four intermediate ANC tests were conducted in a duplicate manner by adding 23 mL, 46 mL, 68 mL, and 92 mL simulated AMD, i.e., 20%, 40%, 60%, and 80% of the added acid amount at the endpoint. For brevity, the four intermediate samples and the endpoint one were denoted as S1, S2, S3, S4, and S5.

#### 3.3.1. Leaching Characteristics

Leaching ratios and concentrations for different samples are shown in Figure 4a,b, and the results indicated that with simulated AMD amount increasing, all components had increased leaching ratios and concentrations except for Ca.

For Ca-bearing components, sample S2 had the maximum leaching ratio and leaching concentration, which were 28.73% and 9.20 mg/L, respectively.

Mg-bearing constitutes was another important alkaline resource, its leaching ratio increased directly with the added simulated AMD increasing. S5 had a maximum value of 56.75% at the endpoint. Yet Mg-bearing ingredients in slags were relatively small, so its final concentration was only 2.03 mg/L.

Si-bearing components in steel slag mainly existed in form of C_2_S, C_3_S, and calcium magnesium silicate (C_2_MS, 2CaO·MgO·SiO_2_), therefore, when the Ca/Mg-bearing silicate components dissolved in solutions, both the Si-bearing and Ca/Mg-bearing components had similar variation trends as illustrated in Figure 4.

From Figure 4, it can be seen that the Al-, Fe- and Cr-bearing constitutes in steel slag had similar variation trends. Their leaching ratios were close to zero until the simulated AMD amount reached 69 mL, which suggested that the Al-, Fe-, and Cr-bearing components were likely to precipitate as hydroxides under alkaline conditions.

#### 3.3.2. Phase Characteristics

Steel slag phase transformation in the neutralization process is shown in Figure 5. Compared with the raw steel slag sample, samples S1 to S5 had stronger diffraction intensity, which may be caused by the removal of the amorphous mineral when the steel slag was washed with the deionized water and acid solution. With the consumed simulated AMD increasing, the CaSO_4_·0.5 H_2_O diffraction intensity became stronger while the raw sample components became weaker. Although the exact disappearance sequence for different mineral phases in steel slags was difficult to detect by XRD analysis, some interesting clues emerged which were highlighted by the dotted line in Figure 5. It is known that the f-CaO was one of the most active alkaline components in steel slags, but it was not prominent in the XRD patterns because of its small mass percentage. Yet for sample S2, there was an obvious f-CaO, which may be due to the dissolution of the outer layer of the f-CaO constitute. Further dissolution reactions make the f-CaO peak disappear again. Another interesting line was located at the CaFeSi_2_O_6_ peaks because it was persistent in the neutralization process in samples S1 to S5. It can be speculated that the CaFeSi_2_O_6_ was not likely to dissolve in simulated AMD. As for Ca_2_SiO_4_ and Ca_3_SiO_5_, the major Ca-bearing components in steel slag, their peaks had different trends by comparison to the diffraction peak in the range of 26.2° and 37.4°. The disappearance of Ca_2_SiO_4_ peaks was accompanied by the formation of CaSO_4_·0.5 H_2_O throughout the neutralization process, while C_3_SiO_5_ peaks disappeared at the beginning period. Therefore, the C_3_SiO_5_ phase was possibly easier to react with the simulated AMD than Ca_2_SiO_4_.

To investigate the phase change for a single component of real steel slags in acid solutions, thermodynamic calculation was used to explore the priorities of different minerals. Possible reactions between slag components and simulated AMD (0.1 M H_2_SO_4_) were listed in Table 2. The related Gibbs free energy changes for the reactions indicated that the reaction sequence of components with simulated AMD was CaO > MgO > FeO > C_2_S > C_2_F > C_3_S under room temperature [12]. However, some synthetic pure Ca- and Mg-bearing minerals studied by leaching experiments and thermodynamic analysis thought that the leaching order for different phases is CaO > (Ca_3_Al_2_O_6_, γ-Ca_2_SiO_4_, Ca_3_MgSi_2_O_8_, and Ca_2_MgSi_2_O_7_) > (Ca_12_Al_14_O_33_, and Ca_2_Fe_2_O_5_) [31].

For the typical components of basic oxygen steel slags, the alkaline components in steel slag could be classified into three groups according to their solubility. The first group was f-CaO, MgO, and C_2_F; the second group included weakly bound CaO, MgO, and some loosed bound C_2_S; while the third group contained some tightly bound C_2_S, C_3_S, MgO, and FeO) [32]. It means that the dissolved CaO, MgO, and FeO were preferential to react with the simulated AMD over C_2_S, C_3_F_2_, and C_3_S [25,32,33].

#### 3.3.3. Morphology Characteristics

Significant changes for slag particle appearance were seen after the acid neutralization process, morphological features of sample S1 to S5 were shown in Figure 6. For samples S1 and S2, most slag particles kept block shapes as raw slags. Yet for samples S3, long trips and well-crystallized particles emerged. As for samples S4 and S5, crystal dimension was much bigger than that in sample S3, in addition, there were some new formed porous precipitate on surface of the crystals. Therefore, with the added simulated AMD increasing, more steel slag ingredients were dissolved and precipitated. The dissolving reaction made the original steel slag particle size became smaller, while the residual was then wrapped by the new formed precipitate and grown bigger.

Chemical compositions for several specific sites on steel slag surfaces of different morphology were tested by EDS, shown in Table 3. Position 1 was selected on the surface of a block porous particle in sample S1, its major elements include calcium, sulfur, and oxygen, and the fewer content elements Si, Al, and Fe were not detected. Its chemical position was not conformed to sulfate calcium or Ca-bearing silicates in raw steel slags. Its high sulfur content indicated the sulfate formation, while the less Si content may be due to the dissolution of the silicates in steel slags. On the basis of the aforementioned phase transition and the micro-structure results, it was speculated that the S1 sample was under the coupled reactions of dissolution and sulfate formation, but the dissolution reaction plays a dominant role. A similar analysis was conducted for position 2 to position 6. Take position 2 for example, its Ca, S, and O percentages were higher than that of position 1, especially since the Si content was very small, thus it may be on the sulfate surface. The high content of Ca and S of position 3 further confirmed the occurrence of sulfate crystal with the simulated AMD increasing. As for position 4, its major elements were Si and O, which may be the leaching residual after Ca- and Mg-bearing components dissolution. Position 5 and position 6 were selected on the surface of well-crystallized sulfate in samples S4 and S5, their main elements were Ca, S, and O, with a minor content of Fe and Si.

Surface area and pore volume determination results were described in Figure 7 and Table 4. The results indicated that the calculated surface area and total volume of steel slag samples increased with the consuming simulated AMD increasing. It is easy to understand that with the dissolution reactions occurring, the Ca- and Mg-bearing constitutes produced some new pores and dissolution channel increasing the surface area and pore diameters. Especially, the overview appearance of the slag particles has changed from block shape into strip shape, suggesting that the new formed sulfate crystal was originated on the original slag surface, then the reactions between the leached calcium and the added sulfate radical making the crystal grown up. Gradually, the slag particles were encased by the new formed sulfate crystals, then the leaching behaviors of the undissolved ingredients in the steel slag were blocked, leading the neutralization reaction to be stopped gradually. In summary, the 40% of the neutralization endpoint was seen to be a turning point for the dissolution reaction and precipitate reaction.

### 3.4. ANC Test by Real AMD

Titration experiments using real AMD were conducted to investigate the effect of components in real AMD on the neutralization property of the slag samples. Compositions of real AMD samples are shown in Table 4.

For acid neutralization capacity tests, titration procedures and endpoint regulation were consistent with the experiments using simulated AMD. Figure 8 illustrates the pH trends in the neutralization process. It can be seen that when the pH reached the endpoint and neutral value (pH = 7), for 1 g steel slag sample, the consumed real AMD was 670 mL and 230 mL, and the calculated ANC was 2.34 mmol H^+^/g slag and 0.85 mmol H^+^/g slag, respectively. In comparison to simulated AMD, the ANC results for real AMD were relatively lower, which may be caused by the metal ions in real AMD, such as Fe^2+^, Mg^2+^, and Al^3+^, because they would also consume the alkaline components in steel slag leaching solutions through hydrolysis reactions and hydroxide formation [14,34,35]. Element distribution on the steel slag surface was confirmed by EDS, the results indicated that the Ca-bearing areas overlapped with the S areas; thus, it can be speculated the sulfate-containing compounds were formed and precipitated during the neutralization process. Moreover, the Fe-, Si-and Mg-enriched areas on the surface were observed at else areas, which may be caused by the hydroxides of the metal ions precipitated (Equation (2)) or adsorbed (Equation (3)) from AMD solution as followings.
> Si - OH··· H- O- H [Me(OH_2_)_3_]^2+^ ⟷ > Si- O Me + H_3_O^+^(2)
Me^2+^ + OH^−^ ⟷ Me(OH)^+^+OH^−^ ⟷ Me(OH)_2_(3)

Combining the obtained microscopic properties and ANC results in the neutralization process of the steel slag samples, the overall reaction between AMD and slag samples could be summarized. In summary, slag leaching and sulfate precipitate were two categories of reactions throughout the neutralization process with 40% of the endpoint simulated AMD volume as a turning point. Although the two kinds of reactions occurred simultaneously, with the solution pH decreased from alkaline to acidic, each component presented diverse changes.

In the initial stage, the slag leaching reaction played a dominant role since f-CaO, MgO, some loosed bound CaO, and C_2_S were released into the solution and hydrated with the acidic component in AMD, which caused Ca concentration to increase and sulfate calcium on the slag surface. As for the slag samples in this work, the easily leaching Ca-bearing constitutes were f-CaO, C_2_S, C_3_S, and CF, among which C_3_S may contribute more to alkaline neutralization capacity, for the f-CaO amount is few and the C_2_S is relatively less active. As the simulated AMD volume continues increasing, the formed sulfate calcium crystal grows up making the original steel slag appearance change into long strip forms by wrapping or connecting steel slag particles. Additionally, the dissolution of the alkaline ingredients makes the samples have a larger pore diameter and surface area, but the channels of the steel slag particles were blocked by the produced sulfate which makes the alkaline releasing behaviors gradually stop. Similar external coating and armors of iron hydroxides were also found when AMD was neutralized by lime in the passive treatment [25,36,37]. For the real AMD titration sample, the precipitate was a mixture of sulfate and metal hydroxides from real AMD, and that is why the real AMD sample had a smaller ANC value.

## 4. Conclusions

AMD remediation by steel slag could make full use of the alkaline substances in steel slag. With the dissolution of slag proceeding, Ca and Mg concentration in the solution increased. The maximum leaching ratio of Ca was about 28.73% when 40% of the alkalinity was neutralized. It can be considered the turning point of the dissolution stage and precipitate stage. The newly formed sulfate crystals on the steel slag surface prevent the inner constitutes in the steel slags from further leaching. The ANC results indicated that there was a linear relationship between pH variation and AMD amount for both simulated and real AMD. When pH decreased to the neutral range, the calculated ANCs were 8 mmol H^+^/g slag and 0.85 mmol H^+^/g slag for simulated AMD and real AMD, respectively. In addition, the particle size distribution had a significant influence on the ANC property of slag. Therefore, the utilization ratio of alkaline substances in steel slag was not high, especially for large particles employed in the AMD treatment application. To further improve the alkaline substance usage rate, the precipitate process on the surface of slag should be inhibited in future investigations.

## Figures and Tables

**Figure 1 ijerph-20-02805-f001:**
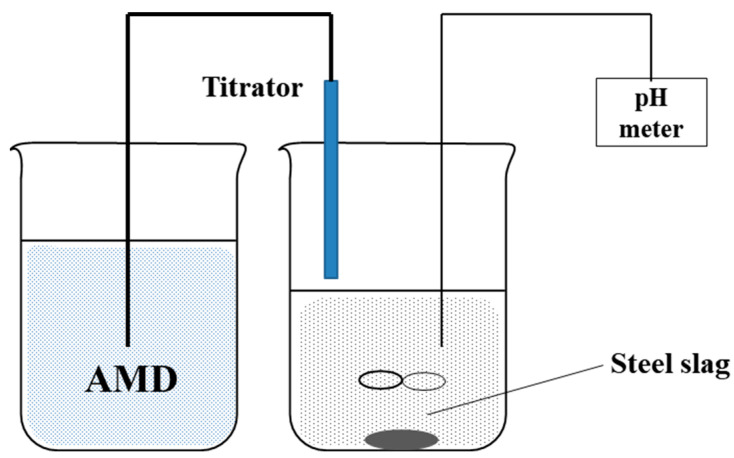
Setup of ANC determination experiment.

**Figure 2 ijerph-20-02805-f002:**
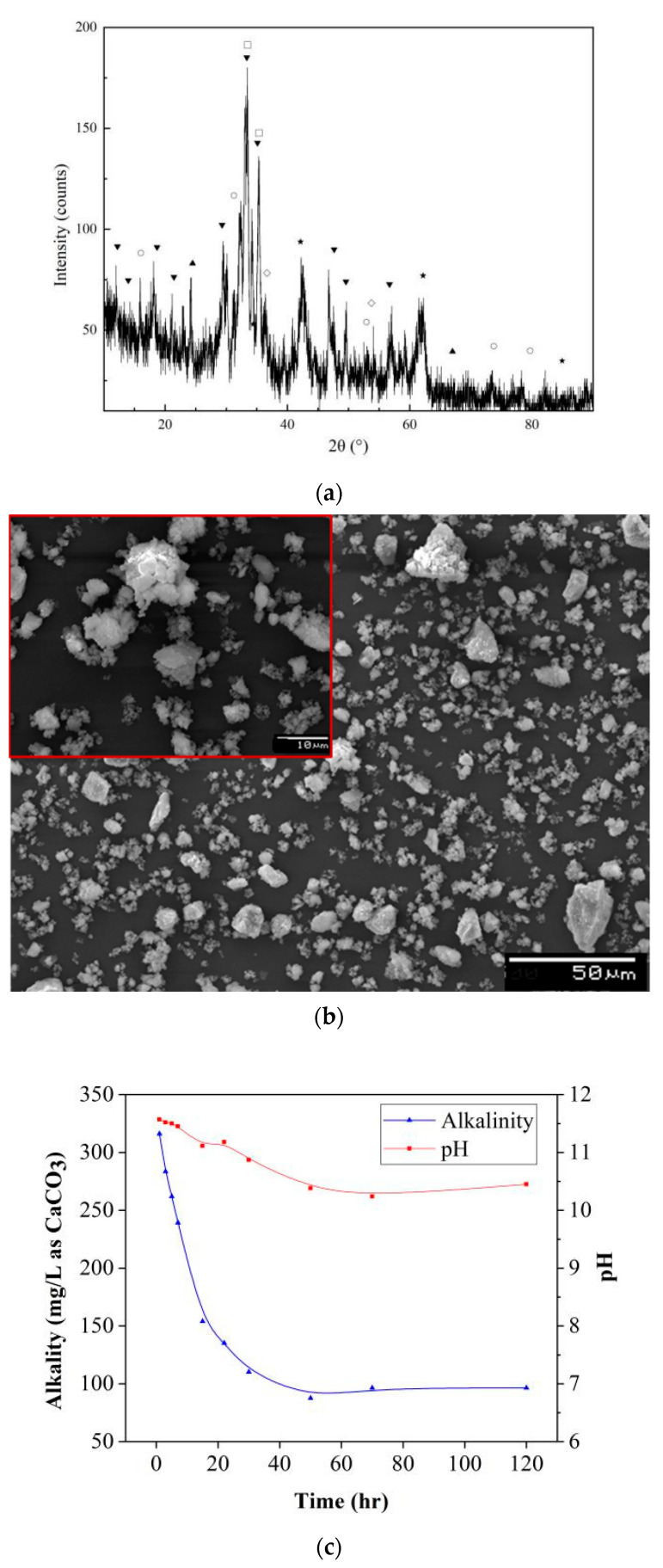
Characterization results of raw steel slag. (**a**) XRD patterns: (○) C_2_F, (▲) Mg-Al hydrotalcite, (◊) f- CaO, (▼) C_2_S, (★) RO, (□) C_3_S; (**b**) SEM image; (**c**) alkalinity production property.

**Figure 3 ijerph-20-02805-f003:**
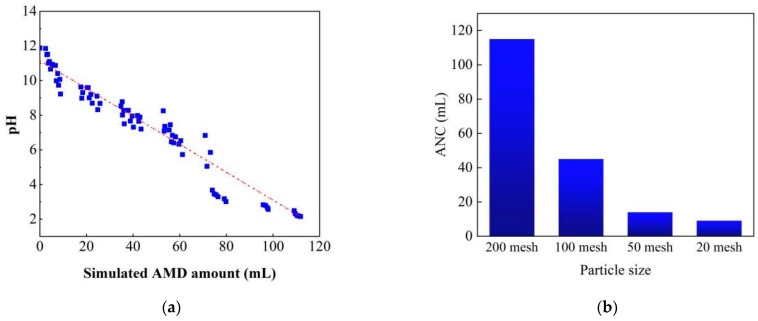
ANC determination results, (**a**) pH variations with simulated AMD; (**b**) influence of particle sizes on slag ANC.

**Figure 4 ijerph-20-02805-f004:**
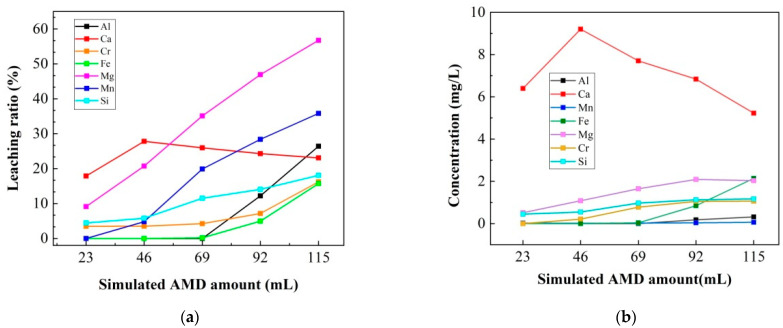
Leaching feature during the neutralization process, (**a**) leaching ratios; (**b**) leaching concentrations. Ca-, Mg-, Si-, Al-bearing components had been studied extensively in regards to their leaching behaviors. It was reported that the Mg-, Ca-bearing constituents in steel slag contribute hugely to the short-term acid neutralization process, while Si/Al-bearing constituents discharge alkaline materials over a longer period of time due to their lower dissolution rate [25]. Slag leaching experiments conducted by HCl (0.1 M) also confirmed that CaO was more active than other Ca-containing minerals, its leaching ratio was more than 80%; followed by was silicates, its leaching ratio was more than 45%; then were the aluminates and ferrite, their leaching ratios were less than 30% [30].

**Figure 5 ijerph-20-02805-f005:**
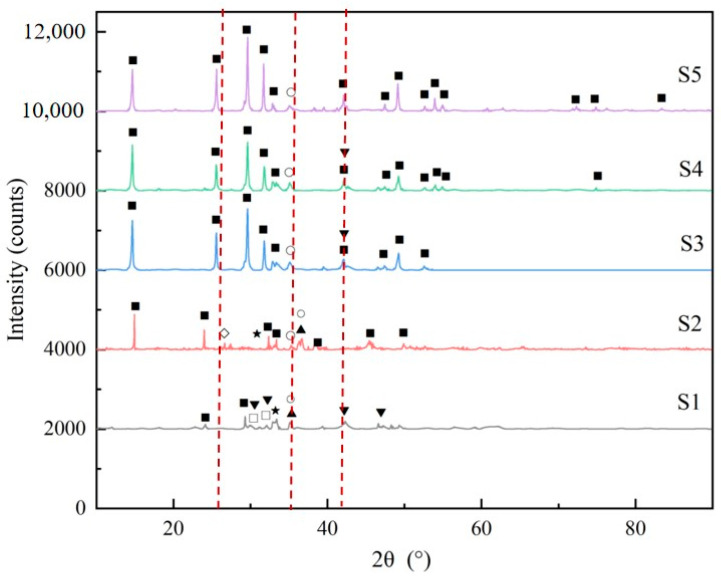
XRD patterns of slag samples titrated by simulated AMD. The following compounds were identified: (■) CaSO_4_·0.5 H_2_O; (○) CaFeSi_2_O_6_; (▲) Mg-Al hydrotalcite; (◊) CaO; (▼) Ca_2_SiO_4_; (★) RO, (Mg, Mn, Ca)_x_ Fe_1−x_O; (□) Ca_3_SiO_5_.

**Figure 6 ijerph-20-02805-f006:**
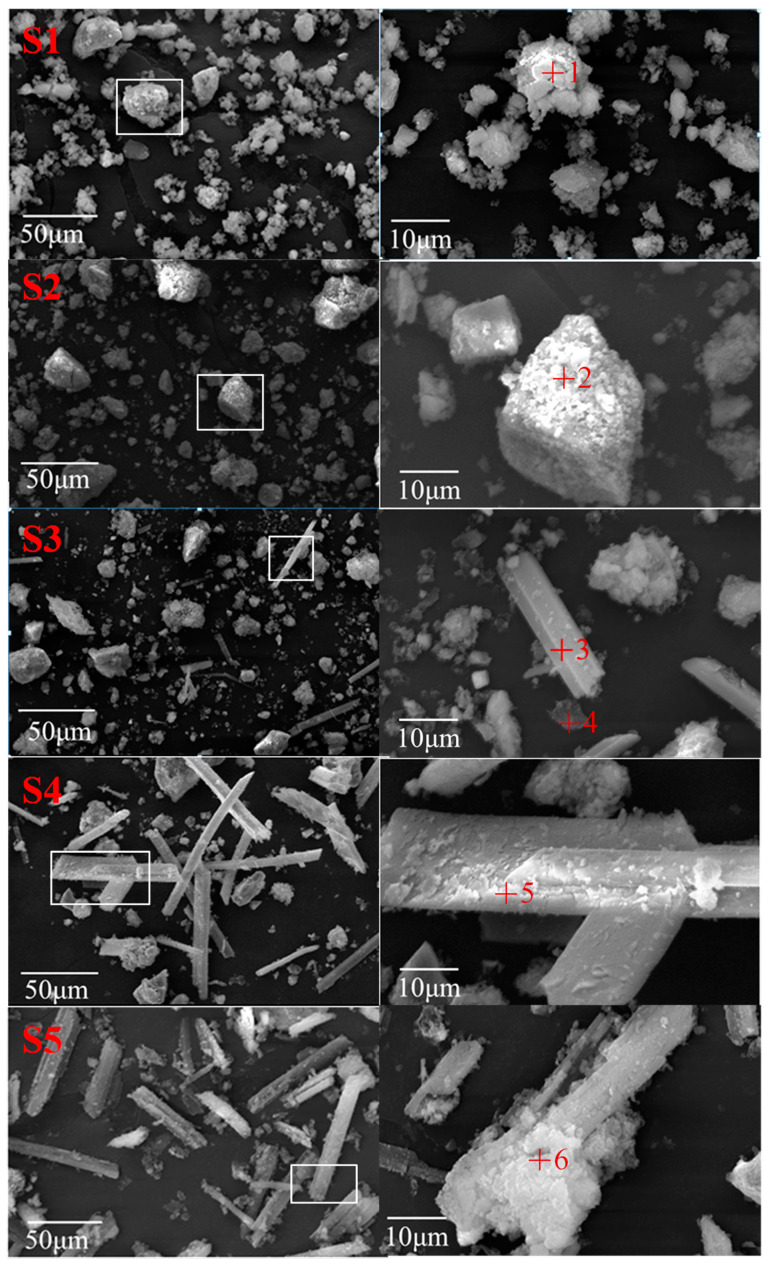
SEM images of steel slags titrated with simulated AMD.

**Figure 7 ijerph-20-02805-f007:**
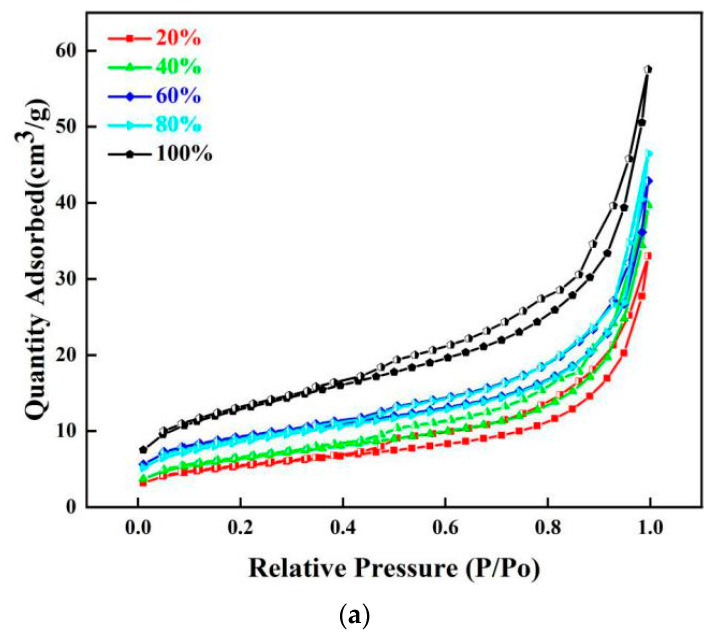
Adsorbed quality and pore volume for simulated AMD titrated slags, (**a**) surface area; (**b**) pore volume, (**c**) BET surface area and total pore volume.

**Figure 8 ijerph-20-02805-f008:**
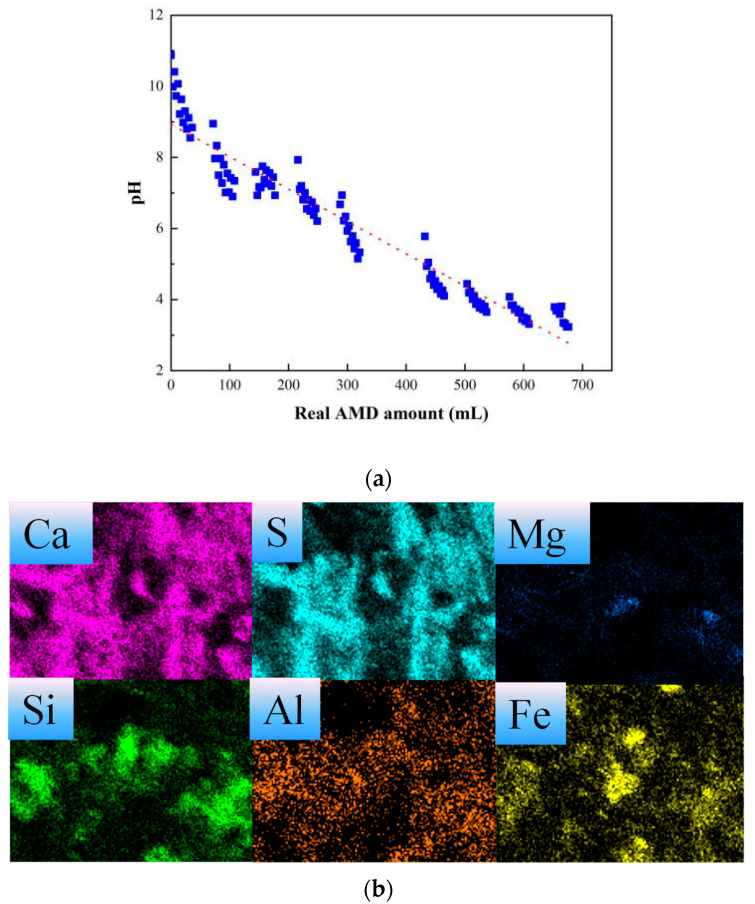
Experimental results for steel slags neutralized by real AMD, (**a**) pH variation, (**b**) element distribution on steel slag surface.

**Table 1 ijerph-20-02805-t001:** Composition of raw steel slags.

Composition	CaO	Fe_2_O_3_	SiO_2_	MgO	MnO	Al_2_O_3_	P_2_O_5_	V_2_O_5_	Cr_2_O_3_
%	41.82	25.05	11.89	6.61	5.48	2.18	1.59	0.73	0.69

**Table 2 ijerph-20-02805-t002:** Possible reactions occurred in the titration process.

**Component**	**Reactions**
CaO	$\mathrm{CaO}+2H^{+}\to {\mathrm{Ca}}^{2+}+H_{2}O$
MgO	$\mathrm{MgO}+2H^{+}\to {\mathrm{Mg}}^{2+}+H_{2}O$
$\mathrm{Ca}{\left(\mathrm{OH}\right)}_{2}$	$\mathrm{Ca}{\left(\mathrm{OH}\right)}_{2}+2H^{+}\to {\mathrm{Ca}}^{2+}+2H_{2}O$
$\mathrm{Mg}{\left(\mathrm{OH}\right)}_{2}$	$\mathrm{Mg}{\left(\mathrm{OH}\right)}_{2}+2H^{+}\to {\mathrm{Mg}}^{2+}+2H_{2}O$
${\mathrm{Ca}}_{2}{\mathrm{SiO}}_{4}$	
	$2\mathrm{CaO}+4H^{+}\to 2{\mathrm{Ca}}^{2+}+2H_{2}O$
${\mathrm{Ca}}_{3}{\mathrm{SiO}}_{5}$	${\mathrm{Ca}}_{3}{\mathrm{SiO}}_{5}\to 3\mathrm{CaO}+S{\mathrm{iO}}_{2}$
	$3\mathrm{CaO}+6H^{+}\to 3{\mathrm{Ca}}^{2+}+3H_{2}O$
	${\mathrm{CaFe}}_{2}O_{5}\to \mathrm{CaO}+{\mathrm{Fe}}_{2}O_{3}$
	$3\mathrm{CaO}+6H^{+}\to 3{\mathrm{Ca}}^{2+}+3H_{2}O$
${\mathrm{CaSO}}_{4}$	${\mathrm{Ca}}^{2+}+{\mathrm{SO}}_{4}^{2-}\to {\mathrm{CaSO}}_{4}$

**Table 3 ijerph-20-02805-t003:** The elemental compositions of the slags via EDS.

Site	Ca	S	O	Mg	Si	Al	Fe
1	16.72	12.12	47.64	0.02	22.7	0.26	0.54
2	24.36	19.74	53.94	0.03	1.88	0.05	-
3	48.05	34.12	17.7	-	0.07	-	-
4	5.8	3.96	40.62	0.02	48.37	0.58	0.58
5	42.12	31.62	26.09	0.04	-	0.04	-
6	57.06	34.4	6.63	-	0.54	-	1.30

**Table 4 ijerph-20-02805-t004:** Compositions of real AMD.

Element	Na	Mg	Al	K	Ca	Mn	Fe	Co	Ni	Zn	SO_4_^2−^
Content (mg/L)	67.34	123.13	43.16	9.80	85.35	1.56	495.82	0.32	0.40	1.17	2689.96

## Data Availability

All data generated or analyzed during this study are included in this published article and its Appendix A.

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
