# Peer review of "Remediation of Acid Mine Drainage (AMD) Using Steel Slag: Mechanism of the Alkalinity Decayed Process"

_ijerph, 2023, doi:10.3390/ijerph20042805_

Round 1
Reviewer 1 Report
This is an article with a lot of information and abundant results that explain the behavior of steel lag when it interacts with AMD.
Although there is a serious drawback: most of the results are derived from experiments carried out with the so-called synthetic AMD: an erroneous and excessive concept since it is a simple solution of dilute sulfuric acid, which bears very little resemblance to real AMD, except for acidity and the presence of sulfate ions. In fact, the results differ significantly when compared to those found using real AMD, as the authors state: “In comparison to simulated AMD, the ANC results for real AMD were relatively lower, which may be caused by the metal ions in real AMD, like Fe2+, Mg2+, Al3+, because they would also consume the alkaline components in steel slag leaching solutions through hydrolysis reactions and hydroxide formation (Sephton and Webb 2019, Tabelin et al. 2020)”.
A synthetic AMD must imitate real AMD and, therefore, must have an acidic pH, high conductivity, highly oxidizing redox potential, and a high concentration of Fe and SO4= (as shown in table 5) while the presence of other elements depend on AMD source. On the other hand, in nature there are not only physical-chemical interactions, but a variety of bacteria have a very active role in oxidation processes (in aerobic conditions) and reduction (in anaerobic conditions), in addition to promoting both bioleaching processes and bioprecipitation depending on media composition and aeration level. It is evident that considering all these variables, the analysis of the damping effect of AMD through the addition of alkaline compounds is much more difficult to analyze.
However, the article may be published because the information provided is relevant with the only caveat that the concept of synthetic AMD should be dropped: it is better to just say a dilute solution of sulfuric acid.
Reviewer 2 Report
Manuscript entitled “Remediation of acid mine drainage (AMD) using steel slag: mechanism of the alkalinity decayed process” submitted by Lei Yang, Yuegang Tang, Duanning Cao, Mingyuan Yang, Yuanyuan He and Cunfang Lu, can be considered for publication in International Journal of Environmental Research and Public Health, after a major revision.
Here is a list of my specific comments:
- Page 1, Abstract: Include in this section the most important experimental results to highlight the importance of this study.
- Page 2, line 47: Table 1 should be moved into Supplementary materials.
- Page 2, line 62: “Previous studies have confirmed…”. Add here more references.
- Page 3, line 98: “Therefore, ANC and other properties…”. At the end of Introduction, the main objectives of this study should be clearly and detailed presented.
- Page 3, Figure 1: This figure should be mentioned in the text.
- Page 4, 3. Results: Replace this title by “3. Results and discussion”, and provide a detailed discussion of the experimental results immediately after their presentation.
- Page 6, line 204: “ANC results for slag sample…”. This paragraph should be clearly reworded.
- Page 7, 3.3.1. Leaching characteristics: The experimental results included in this section should be clearly presented and discussed.
- Page 16, 4. ANC consumption mechanism and discussion: This title should be deleted, and the observations moved in the previous sections.
- Page 17, Reference: The number of references should be increased.
Author Response
Authors’ responses to reviewers’ comments
The authors wish to begin with thanking all the reviewers for their constructive criticism, which have significantly improved the manuscript. Their comments can be seen in the following section, as well as the authors’ point-to-point reply in blue.
Reviewers' comments:
Reviewer #2:
Q1 Page 1, Abstract: Include in this section the most important experimental results to highlight the importance of this study.
A1 Thanks for the valuable comment from the reviewer. As suggested by the reviewer,the ANC results was added into the abstract as following.
(page 1, line 148-150)
For steel slag of 200 mesh size, the ANC value for steel slag sample were 8.23 mmol H+/g when dilute sulfate acid was used.
Q2 Page 2, line 47: Table 1 should be moved into Supplementary materials.
A2 Thanks for the reviewer’s comments. As the reviewer suggested, Table 1 was moved into Supplementary materials. And the number of the following tables were reordered.
(Page 2, line 48)
Table 1 was replaced by S1.
Q3 Page 2, line 62: “Previous studies have confirmed…”. Add here more references.
A3 Thanks for the reviewer’s comment. As the reviewer suggested, more references about alkalinity production of steel slag were added.
(Page 2, line 64-66)
Alakangas, Andersson, Mueller (2013) Neutralization/prevention of acid rock drainage using mixtures of alkaline by-products and sulfidic mine wastes. Environ Sci Pollut Res Int, 20: 7907-7916.https://doi:10.1007/s11356-013-1838-z
Gomes, Mayes, Baxter, et al. (2018) Options for managing alkaline steel slag leachate: A life cycle assessment. Journal of Cleaner Production, 202: 401-412.https://doi:10.1016/j.jclepro.2018.08.163
Riley, Mayes (2015) Long-term evolution of highly alkaline steel slag drainage waters. Environ Monit Assess, 187: 463.https://doi:10.1007/s10661-015-4693-1
Q4 Page 3, line 98: “Therefore, ANC and other properties…”. At the end of Introduction, the main objectives of this study should be clearly and detailed presented.
A4 Thanks for the reviewer’s comment. As the reviewer suggested, the objective of the study was added at the end of the introduction.
(Page 3, line 101-103)
The main objective of the manuscript was the study the decay mechanism of the acid neutralization capacity for the steel slag during the acid mine drainage remediation process.
Q5 Page 3, Figure 1: This figure should be mentioned in the text.
A5 Thanks for the reviewer’s comment. Figure 1 was mentioned and described in the manuscript.
(Page 3, line 130)
Q6 Page 4, 3. Results: Replace this title by “3. Results and discussion”, and provide a detailed discussion of the experimental results immediately after their presentation.
A6 Thanks for the reviewer’s comment. The title Results and discussion was replaced and the detailed discussion the experimental results were added.
Q7 Page 6, line 204: “ANC results for slag sample…”. This paragraph should be clearly reworded.
A7 Thanks for the reviewer’s comment. As suggested by the reviewer, the ANC results for slag sample was reworded more clearly.
(Page 6, lines 201-212)
Experimental results of acid neutralization capacity test by simulated AMD (0.1 M H2SO4)was presented in Fig.3 (a). It can be seen that with the simulated AMD amount increasing, pH of the solution decreased directly from initial 10.5 to 2.15 at the endpoint gradually. The total consumed volume of the simulated AMD was 115 mL for 1 g steel slag samples with average particle size of of 200 mesh. And it was about 40 mL when the pH reached the neutral value of 7. The calculated ANC values was 8.23 mmol H+/g for the steel slag sample, which was agreed well with the reported ANC results of electric arc furnace steel slag. The electric arc furnace steel slag was neutralized by similar method using HNO3 (1M) as simulated AMD, when pH decreased to 7 (Yan et al. 2000), the calculated ANC of EAF slag was about 5.5 mmol H+/g slag and 7.5 mmol H+/g slag for short and long experiments respectively.
Q8 Page 7, 3.3.1. Leaching characteristics: The experimental results included in this section should be clearly presented and discussed.
A8 Thanks for the reviewer’s comment. As suggested by the reviewer, the leaching characteristics of steel slag were discussed again.
(Page 8, lines 253-267)
Leaching ratios and concentrations for different samples were shown in Fig. 4 (a) and Fig. 4 (b) the results indicated that with simulated AMD amount increasing, all components had increased leaching ratios and concentrations except for Ca..
For Ca-bearing components, sample S2 had the maximum leaching ratio and leaching concentration, which were 28.73 %and 9.20 mg/L, respectively.
Mg-bearing constitutes was another important alkaline resource, its leaching ratio increased directly with the adding simulated AMD increasing. S5 had the maximum value of 56.75 % at endpoint. But Mg-bearing ingredients in slags were relatively small, so its final concentration was only 2.03 mg/L.
Si-bearing components in steel slag were mainly existed in form of C2S, C3S and calcium magnesium silicate (C2MS, 2CaO·MgO·SiO2), therefore, when the Ca/Mg-bearing silicate components were dissolved in solutions, both the Si-bearing and Ca/Mg-bearing components had similar variation trends as illustrated in Fig. 4.
Ca, Mg, Si, Al-bearing components had been studied extensively in regards to their leaching behaviors. It was reported that the Mg-,Ca-bearing constituents in steel slag contribute hugely to the short-term acid neutralization process, while Si/Al-bearing constituents discharge alkaline materials over a longer period of time due to their lower dissolution rate (Manchisi et al. 2020). Slag leaching experiments conducted by HCl (0.1M) also confirmed that CaO was more active than other Ca-containing minerals, its leaching ratio was more than 80%; followed by was silicates, its leaching ratio was more than 45%; then were the aluminates and ferrite, their leaching ratios were less than 30 % (Hall et al. 2014).
Q9 Page 16, 4. ANC consumption mechanism and discussion: This title should be deleted, and the observations moved in the previous sections.
A9 Thanks for the reviewer’s comment. The title ”ANC consumption mechanism and discussion”was deleted and the description was incorporated into Chapter 3.
Q10 Page 17, Reference: The number of references should be increased.
A10 Thanks for the reviewer’s comment. As suggested by the reviewer, the number of the reference increased , and the increased following references were also cited in the manuscript.
Gao, Jiang, Tian, et al. (2017) BOF steel slag as a low-cost sorbent for vanadium (V) removal from soil washing effluent. Sci Rep, 7: 11177.https://doi:10.1038/s41598-017-11682-3
Kruse, Hawkins, López, et al. (2019) Recovery of an Acid Mine Drainage-Impacted Stream Treated by Steel Slag Leach Beds. Mine Water and the Environment, 38: 718-734.https://doi:10.1007/s10230-019-00636-y
Masindi, Osman, Abu-Mahfouz (2017) Integrated treatment of acid mine drainage using BOF slag, lime/soda ash and reverse osmosis (RO): Implication for the production of drinking water. Desalination, 424: 45-52.https://doi:10.1016/j.desal.2017.10.002
Masindi, Osman, Mbhele, et al. (2018) Fate of pollutants post treatment of acid mine drainage with basic oxygen furnace slag: Validation of experimental results with a geochemical model. Journal of Cleaner Production, 172: 2899-2909.https://doi:10.1016/j.jclepro.2017.11.124
Ntuli, Magwa (2018) Sulphate removal from acid rock drainage using steel slag. IOP Conference Series: Earth and Environmental Science, 191: 012116.https://doi:10.1088/1755-1315/191/1/012116
Round 2
Reviewer 2 Report
Manuscript entitled “Remediation of acid mine drainage (AMD) using steel slag: mechanism of the alkalinity decayed process” submitted by Lei Yang, Yuegang Tang, Duanning Cao, Mingyuan Yang, Yuanyuan He and Cunfang Lu, can be considered for publication in International Journal of Environmental Research and Public Health, after a minor revision.
Here is a list of my specific comments:
- Page 4, line 162: “the main objectives of this…”. This sentence should be deleted.
- Page 6, line 199: “…pH of the solution decreased directly…”. This variation should be explained.
- Page 7, line 236: “The experimental results included…”. This should be deleted.
- Page 10, line 333: “But some synthetic pure…”. What is the importance of this observation??? Please explain this.
Author Response
Authors’ responses to reviewers’ comments
The authors wish to begin with thanking all the reviewers for their constructive criticism. The comments can be seen in the following section, as well as the authors’ point-to-point reply in blue.
Q1. Page 4, line 162: “the main objectives of this…”. This sentence should be deleted.
A1 Thanks for the reviewer’s valuable comments. The sentence was deleted.
Page4 line 158
The main objectives of the ... detailed presented.
Q2. Page 6, line 199: “…pH of the solution decreased directly…”. This variation should be explained.
A2 Thanks for the reviewer’s comments. As suggested by the reviewer, the reason of the pH variation was insert in the manuscript.
Page 6 line 196-200.
It is known that the alkaline ingredients in steel slag could leach into the solution, raising the solution pH to alkaline range. For 1 g slag sample, the final pH was 10.5. While with the added dilution sulfuric acid amount increasing, the produced OH- was neutralized, which make the pH increased slowly although the leaching reaction was going on.
Q3 Page 7, line 236: “The experimental results included…”. This should be deleted.
A3 Thanks for the reviewer’s comments. The sentence was deleted.
Page 7 line 236.
Q4 Page 10, line 333: “But some synthetic pure…”. What is the importance of this observation??? Please explain this.
A4 Thanks for the reviewer’s comments. This was cited to discuss the priority of the steel slag ingredients. Comparing the experimental results from real steel slag and the pure synthetic Ca and Mg bearing minerals, the leaching order was different. It had a significant influence on mineral variation during the neutralization process.